# Exploring the Potential of a P2X3 Receptor Antagonist: Gefapixant in the Management of Persistent Cough Associated with Interstitial Lung Disease

**DOI:** 10.3390/medicina61050892

**Published:** 2025-05-14

**Authors:** Tomoyuki Takahashi, Atsushi Saito, Takafumi Yorozuya, Hirotaka Nishikiori, Koji Kuronuma, Hirofumi Chiba

**Affiliations:** Department of Respiratory Medicine and Allergology, School of Medicine, Sapporo Medical University, Sapporo 060-8543, Japan; tomorrow9900@gmail.com (T.T.);

**Keywords:** cough, gefapixant, P2X3 receptor, interstitial lung disease

## Abstract

*Background:* Interstitial lung disease (ILD) is characterized by pulmonary inflammation and fibrosis associated with persistent and refractory cough that significantly hinders quality of life. Conventional treatments for ILD-associated cough have shown limited efficacy, necessitating alternative therapeutic approaches. Gefapixant, a P2X3 receptor antagonist, can potentially alleviate chronic cough by inhibiting the ATP-mediated activation of sensory C-fibers, but its efficacy in ILD-associated cough remains unclear. This study observed the effects of gefapixant on ILD-associated refractory chronic cough. *Methods:* This prospective study enrolled patients with ILD-associated refractory chronic cough who received gefapixant at Sapporo Medical University Hospital between July 2022 and November 2023. Cough frequency, Leicester Cough Questionnaire (LCQ) score, cough severity visual analog scale (Cough VAS), and taste VAS were evaluated at baseline and at 2, 4, and 8 weeks after gefapixant administration. *Results:* Six patients completed the study. Their ILD subtypes included idiopathic pulmonary fibrosis (IPF), nonspecific interstitial pneumonia (NSIP), and connective tissue disease-associated ILDs (CTD-ILDs). After 8 weeks, the cough frequency decreased from 88.5 to 44.3 episodes per 30 min, LCQ scores increased from 8.3 to 13.6, and cough VAS scores decreased from 75.8 to 40.2. However, statistical significance was not reached due to high interindividual variability, with gefapixant being effective in some and ineffective in others. The most common adverse event was taste disorder, leading to discontinuation in one patient, but symptoms tended to lessen over the course of treatment. *Conclusions:* Gefapixant appears to be effective in reducing refractory cough related to ILD, although these results were not statistically significant because its effectivity widely varied across individuals. Further investigation is needed to identify patient subgroups with the greatest potential for treatment responsiveness.

## 1. Introduction

Interstitial lung disease (ILD) presents with pulmonary disorders of inflammation and fibrosis [1], which is commonly associated with chronic cough that can impair quality of life, among other symptoms [2]. In clinical practice, conventional therapeutic interventions often fail to adequately alleviate this distressing symptom of cough [3]. Although the mechanisms of ILD-associated cough are not fully understood, it is hypothesized that the traction force from pulmonary fibrosis stimulates afferent nerve receptors or destroys nerves that inhibit cough [4]. Another suggested mechanism involves the elevated ATP levels found in the airways in ILD. ATP is released from epithelial cells and binds to P2X3 receptors on C-fibers in the vagus nerve, where it is sensed as a C fiber damage signal and triggers the cough reflex [5].

Gefapixant, a P2X3 receptor antagonist, belongs to a new class of antitussives for refractory or unexplained chronic cough. It inhibits sensory nerve activation by blocking extracellular ATP signaling through its antagonistic effects on P2X3 receptors found on the sensory C-fibers of the vagus nerve [6,7]. Large randomized controlled trials have demonstrated the efficacy of gefapixant for refractory or unexplained chronic cough, but these trials excluded participants with abnormal chest imaging findings. Previous studies also suggest that gefapixant may be useful for refractory cough in idiopathic pulmonary fibrosis (IPF) [8]. One clinical trial in patients with IPF did not show a reduction in cough frequency, the primary endpoint, but secondary endpoints and post hoc analyses suggested a potential therapeutic effect [8].

Considering the potential of this new drug, this study aimed to evaluate the efficacy of gefapixant in the treatment of refractory cough in ILDs in clinical practice.

## 2. Methods

### 2.1. Study Design and Settings/Participants

This study enrolled patients with refractory chronic cough who were prescribed gefapixant at Sapporo Medical University Hospital between July 2022 and November 2023. All patients had a guideline-based diagnosis of cough caused by ILDs [9,10]. The inclusion criteria were as follows: diagnosis of ILD based on guidelines by multiple respiratory specialists [11,12,13], at least 18 years of age, and chronic cough > 1 year. Patients with a history of hypersensitivity to gefapixant or who were unable to complete the questionnaire were excluded.

### 2.2. Primary and Secondary Endpoints

The primary endpoint was the change in cough frequency from baseline to 8 weeks, and the secondary endpoints were the changes in Leicester Cough Questionnaire (LCQ) score and cough severity visual analog scale (Cough VAS) from baseline to 8 weeks.

### 2.3. LCQ Score, Cough VAS, and Cough Frequency

Cough frequency, LCQ score, and cough VAS were measured at the start of gefapixant treatment and after 2, 4, and 8 weeks during outpatient visits. Cough frequency was measured for 30 min using an IC voice recorder (OLYMPUS Inc., Tokyo, Japan) and was manually counted by respiratory physicians.

### 2.4. Statistical Analysis

Based on previous studies that reported approximately 60% reduction in cough frequency with a standard deviation of 50%, the required sample size was estimated assuming a two-sided α of 0.05 and 80% power. The minimum sample size was six patients. Unpaired *t*-test with Welch correction was used to compare the two groups. A one-way analysis of variance (ANOVA) with post hoc tests for multiple comparisons was used to compare more than three groups. A *p* value < 0.05 was considered statistically significant. Data analysis and graph generation were performed using the GraphPad Prism v9 software (GraphPad, Inc., San Diego, CA, USA).

### 2.5. Ethics Approval and Consent to Participate

The study was approved by the Institutional Review Board of Sapporo Medical University (research study number 342-56). Written informed consent was obtained from all participants, ensuring that they were fully aware of the nature and purpose of the research and had given their voluntary and informed consent to participate.

## 3. Results

### 3.1. Patient Characteristics

Out of eight patients who were initially enrolled, two dropped out from the study due to taste disorder and a change of physician. Table 1 shows the characteristics of the remaining six patients evaluated in this study; 83.3% were female, with a mean age of 65.6 years. The specific types of ILD diagnosed were IPF, nonspecific interstitial pneumonia (NSIP), and connective tissue disease-associated ILD (CTD-ILD) (*n* = 2 each). The identified comorbidities were bronchial asthma, chronic obstructive pulmonary disease, and allergic rhinitis (*n* = 1 each); none had gastrointestinal reflux disease. Four patients used home oxygen therapy. The concomitant medications included central antitussives (codeine phosphate; *n* = 6), oral corticosteroids (*n* = 3), proton pump inhibitors (*n* = 3), inhaled corticosteroids (*n* = 2), antihistamines (*n* = 3), and antifibrotic agents (nintedanib; *n* = 3). No other treatment interventions were introduced for cough and comorbidities aside from the addition of gefapixant during the study period.

### 3.2. Gefapixant Tends to Suppress ILD-Associated Chronic Cough

After 8 weeks of treatment compared to baseline, cough frequency decreased from 88.5 to 44.3 per 30 min, cough VAS scores decreased from 75.8 to 40.2, and LCQ scores increased from 8.3 to 13.6. However, none of the endpoints exhibited significant differences (Figure 1A–C). Patients were divided into two groups based on LCQ score improvement: those who showed improvement with time (red circle) and those who did not improve (blue circle) (Figure 2A). Interestingly, the two groups were clearly divided based on significant differences in response to gefapixant (Figure 2B,C). On further comparison of LCQ score improvement rates (subjective cough evaluation) and cough frequency (objective cough evaluation), some patients had improvements in both, some improved in cough frequency but not in LCQ score, and some did not show improvement of both parameters (Figure 2D). However, there was no clear association between these responses and patient background or ILD type. Meanwhile, taste VAS scores decreased from 100 at baseline to 35.3 at 2 weeks, then increased again to 64.3 and 68.5 at 4 and 8 weeks, respectively (Figure 3).

## 4. Discussion

This study assessed the effectiveness of gefapixant, a P2X3 receptor antagonist, in treating ILD-associated refractory chronic cough. There were trends toward improvement in both the objective endpoint of cough frequency and the subjective endpoints of LCQ score and cough VAS. However, these results were not statistically significant despite the overall impression that gefapixant was effective. Upon further analyzing the individual cases and patient backgrounds, there were very large individual differences in the effectiveness of gefapixant (Figure 2), but the factors behind these variations in response remained unclear. A previous study found that the efficacy of gefapixant for chronic cough also varied greatly between patients [14], similar to our study. Thus, further research is needed to determine the specific phenotypes wherein the drug is effective. Two RCTs have demonstrated the efficacy of gefapixant in refractory or unexplained cough but excluded abnormalities on chest radiographs or CT and did not test efficacy in ILDs [14]. Another RCT of patients with IPF showed no significant difference in the primary endpoint of cough frequency [8]. However, a log-transformed post hoc analysis showed a significant reduction in cough frequency, while a secondary endpoint of the responder analysis showed an improvement in the cough frequency, suggesting efficacy. Although the number of patients in this study was small and did not reach statistical significance, it is likely that similar results to those of these clinical trials would be obtained in real clinical cases. Furthermore, some patients did not have subjective improvement even if their cough frequency decreased. This may indicate the difficulty of subjective assessment in cough studies.

Taste disorder was the most important side effect of gefapixant in this study, similar to a previous study [14]. One patient in our cohort dropped out due to this side effect before completing the study period. However, as shown in the taste VAS results in Figure 3, the degree of taste disorder decreased to less than half of the pre-treatment level after 2 weeks but improved by 8 weeks, suggesting a tendency toward habituation. Similar to the effect on cough, individual differences may have a strong influence on the effect on taste (Figure 2). There was no clear association among taste VAS score, cough frequency, LCQ score, or cough VAS score. Furthermore, there was an impression that the intensity of the subjective individual cough may be related to the continuation of gefapixant from a clinical point of view.

The mechanism of gefapixant, which involves the antagonism of P2X3 receptors on sensory C-fibers of the vagus nerve, is a potentially novel approach for managing cough in ILD, where conventional therapies often fail. Additionally, this mechanism aligns with the theory that elevated ATP levels in the airways may trigger cough reflexes through these receptors [5]. However, the large individual variability in response observed in our study highlights the complex pathophysiology of ILD-associated cough (Figure 2). Future research should aim to identify the predictors of responsiveness to gefapixant, such as genetic markers or specific disease phenotypes.

The limitations of our study include the small sample size and lack of a placebo control, which restrict the generalizability of our results. In addition, some results did not reach statistical significance and only showed trends, again likely because of the limited number of patients. Larger placebo-controlled trials could offer more definitive evidence regarding the efficacy and safety of gefapixant in patients with ILD. Moreover, cough frequency was measured for only 30 min and not the entire day. Therefore, 24 h monitoring, including sleep, would be necessary in the future. Given the inclusion of only Japanese subjects this study, differences in ethnicity, lifestyle, and other background characteristics preclude generalizability of our results. Nevertheless, our findings have important clinical implications. If gefapixant can effectively reduce cough in certain patients with ILD, this could represent a significant advancement in the management of a symptom that markedly impairs quality of life. Additionally, some patients did not respond to gefapixant and demonstrated side effects such as taste disorder, suggesting that careful patient selection and management will be crucial for its clinical use.

## 5. Conclusions

Despite the small sample size and lack of a placebo control in this study, our results suggest the potential of gefapixant in treating ILD-associated refractory cough. These findings also highlight the complexity of treating this condition and underscore the need for further investigation into the mechanisms of gefapixant and its role within the broader therapeutic landscape of ILD.

## Figures and Tables

**Figure 1 medicina-61-00892-f001:**
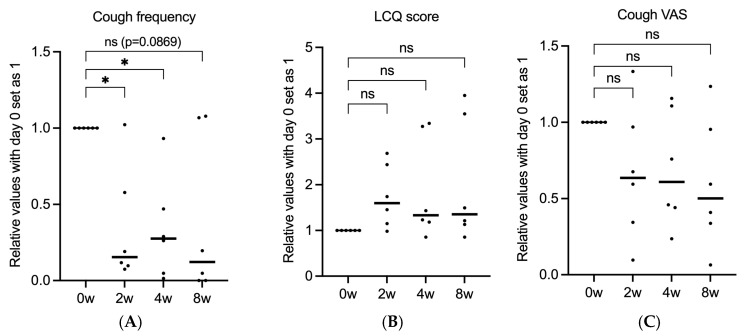
Changes in cough-related parameters over time. The graphs depict the relative values of three different measures with day 0 set as 1: (**A**) cough frequency, (**B**) LCQ score, and (**C**) cough VAS score. Each dot represents an individual data point, and horizontal bars indicate the mean values. Statistical analysis was performed using one-way ANOVA, followed by post hoc tests for multiple comparisons. *: *p* < 0.05, ns: not significant.

**Figure 2 medicina-61-00892-f002:**
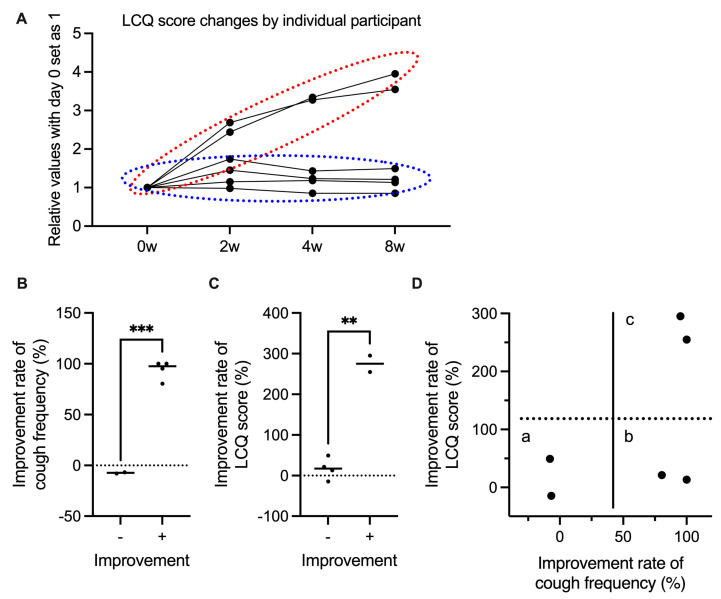
Evaluation of LCQ score and cough frequency by individual participant. (**A**) Changes in LCQ scores over time. Each line represents data from one participant, and the relative values were normalized with day 0 as 1. Participants were divided into two groups based on whether gefapixant was effective (red dashed line) or ineffective (blue dashed line). Improvement rates in (**B**) cough frequency and (**C**) LCQ scores. The two groups were compared using unpaired *t*-test with Welch correction ** *p* < 0.01, *** *p* < 0.001. (**D**) Scatter plot of the relationship between improvement rate in cough frequency and LCQ score. Participants were categorized into three groups: (a) no improvement in both parameters, (b) significant improvement in cough frequency but low LCQ score improvement, and (c) significant improvement in both parameters.

**Figure 3 medicina-61-00892-f003:**
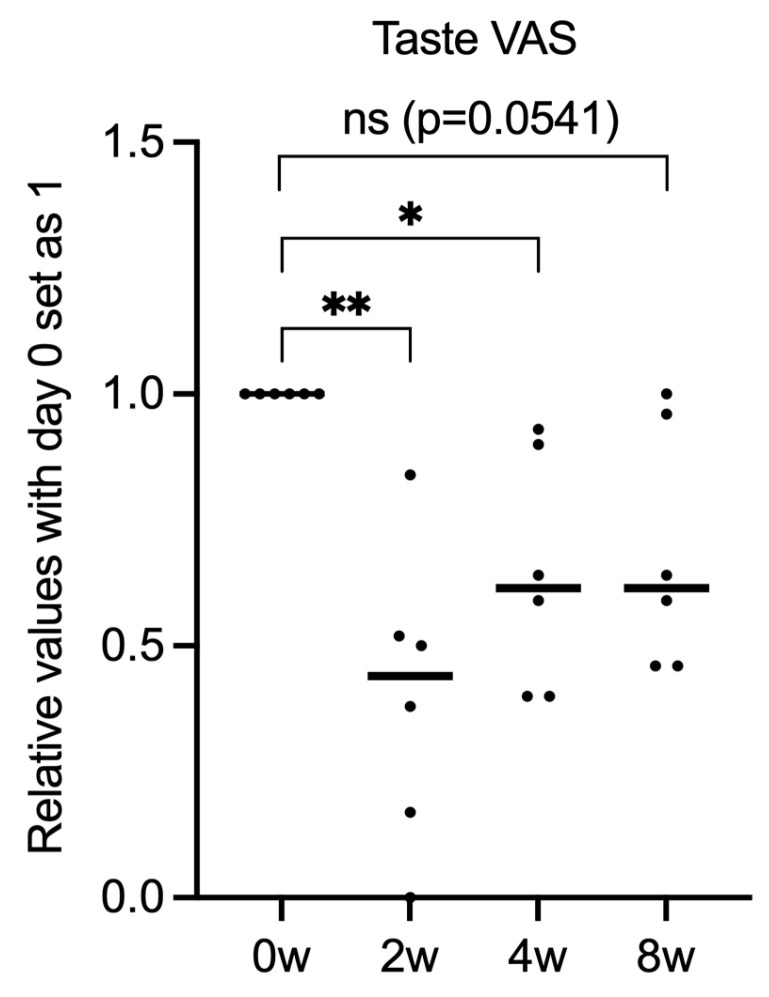
Changes in taste VAS scores over time. The graph depicts the relative values with day 0 set as 1. Each dot represents an individual data point, and horizontal bars indicate the mean values. Statistical analysis was performed using one-way ANOVA, followed by post hoc tests for multiple comparisons. *: *p* < 0.05, **: *p* < 0.01.

**Table 1 medicina-61-00892-t001:** Baseline characteristics of the study participants. Data are presented as N (%) for categorical variables and mean ± SD for continuous variables.

	N	%
Sex		
Male	1	83.3
Female	5	16.7
Age (years)	65.3 ± 12.3	
BMI	26.4 ± 4.3	
Interstitial lung disease		
IPF	2	33
NSIP	2	33
CTD-ILD	2	33
Comorbidities		
Allergic rhinitis	1	16.7
Bronchial asthma	1	16.7
Chronic obstructive pulmonary disease	1	16.7
Gastroesophageal reflux disease	0	0
Concomitant medications		
Narcotic antitussive	6	100
Systemic corticosteroid	3	50
Drugs for obstructive airways disease	2	33
Antihistamine	3	50
Home oxygen therapy	4	66
Baseline values for efficacy endpoints		
Cough frequency (30 min)	88.5 ± 48.7	
ICQ score (baseline total score)	8.3	3.1
Cough VAS (mm)	75.8 ± 16.4	

## Data Availability

Data are available from the author upon request.

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
