# Peer review of "Exploring the Potential of a P2X3 Receptor Antagonist: Gefapixant in the Management of Persistent Cough Associated with Interstitial Lung Disease"

_medicina, 2025, doi:10.3390/medicina61050892_

Round 1
Reviewer 1 Report
Comments and Suggestions for Authors
- The manuscript is written well. However, a few grammatical and punctuation errors should be removed.
- The author should provide the mechanism of action of the Gefapixant, if possible for more clarity. Overall, the manuscript is well explained and presented.
A few grammatical and punctuation errors should be removed.
Author Response
- The manuscript is written well. However, a few grammatical and punctuation errors should be removed.
Response: Thank you for your comment regarding the English language quality of the manuscript. In response to your suggestion, we have resubmitted the revised manuscript for professional English editing by Enago.
- The author should provide the mechanism of action of the Gefapixant, if possible for more clarity. Overall, the manuscript is well explained and presented.
Response: We have described the mechanism of gefapixant in the Introduction, although briefly. As you pointed out, we added some references to clearly describe the mechanism of gefapixant, so that readers can easily understand it. (page 3, line 15)
Reviewer 2 Report
Comments and Suggestions for Authors
I read this paper with great interest. This is a clinically relevant topic, as chronic cough significantly impacts the quality of life in ILD patients, and therapeutic options remain limited. The article is well-written, and the study design is clearly described.
However, I have some major comments:
-There is no mention of sample size calculation or power analysis. Please comment on that.
-The absence of a placebo or comparator group is a strong limitation. Please better address this in discussion.
- Why did you use unpaired t-tests to compare pre- and post-treatment values, given that these are repeated measures from the same individuals?
-Although trends in favor of gefapixant are noted, none of the endpoints achieved statistical significance. The discussion tends to overstate the treatment effect without sufficient caution. Please tune it down.
- Cough frequency was measured only for 30 minutes during the day. This likely underestimates true cough burden and introduces variability. Please address this limitation in more depth.
- Your conclusions are only on Japanese Population. Please comment on that.
- I found several English errors throughout the manuscript. Please have a deep language revision.
- Please Define all acronyms (e.g., LCQ, VAS) at first use in the abstract and main text.
Comments on the Quality of English LanguageModerate revision required
Author Response
I read this paper with great interest. This is a clinically relevant topic, as chronic cough significantly impacts the quality of life in ILD patients, and therapeutic options remain limited. The article is well-written, and the study design is clearly described.
However, I have some major comments:
-There is no mention of sample size calculation or power analysis. Please comment on that.
Thank you for your suggestion. Based on previous studies that reported approximately 60% reduction in cough frequency with standard deviation of 50%, the required sample size was estimated assuming a two-sided α of 0.05 and 80% power. The minimum sample size was calculated to be six patients. These details were added to the Statistical analysis section (page 6, line 20).
-The absence of a placebo or comparator group is a strong limitation. Please better address this in discussion.
Response: Thank you for this suggestion. As pointed out, we discussed this issue in the Limitations section. (page 8, line 9)
- Why did you use unpaired t-tests to compare pre- and post-treatment values, given that these are repeated measures from the same individuals?
Response: Thank you for your comment. We confirm that the individual values shown in Figure 2B represent the percent change in cough frequency for each patient and were calculated using paired measurements before and after treatment. Although calculation of change used paired data, subsequent comparisons between the two subgroups (responders vs. nonresponders) were based on independent observations; therefore, we used unpaired test (Mann–Whitney U test) for this analysis.
-Although trends in favor of gefapixant are noted, none of the endpoints achieved statistical significance. The discussion tends to overstate the treatment effect without sufficient caution. Please tune it down.
- Cough frequency was measured only for 30 minutes during the day. This likely underestimates true cough burden and introduces variability. Please address this limitation in more depth.
- Your conclusions are only on Japanese Population. Please comment on that.
Response: Thank you for this suggestion. As pointed out, these three points were discussed in the Limitations section. (page 8, lines 10-17)
- I found several English errors throughout the manuscript. Please have a deep language revision.
Response: Thank you for your comment regarding the English language quality of the manuscript. In response to your suggestion, we have resubmitted the revised manuscript for professional English editing by Enago.
- Please Define all acronyms (e.g., LCQ, VAS) at first use in the abstract and main text.
Response: Thank you for this suggestion. As pointed out, we have revised accordingly. (page 4, line 12)
Reviewer 3 Report
Comments and Suggestions for Authors
This paper by Takahashi et al. explores the use of gefapixant, a P2X3 receptor antagonist, for treating persistent cough in patients with interstitial lung disease (ILD). Below is a section-by-section analysis of the paper's strengths and weaknesses.
## Introduction
**Strengths:**
- Clearly establishes the clinical problem: ILD-associated cough significantly impacts quality of life and conventional treatments have limited efficacy.
- Provides a sound theoretical basis for using gefapixant by explaining the ATP-mediated mechanism of cough in ILD patients.
- Effectively references previous research on gefapixant in chronic cough management, including its limitations in prior studies.
**Weaknesses:**
- Could benefit from more background on the prevalence and burden of ILD-associated cough to better establish the significance of the research.
- Limited discussion of alternative treatment approaches currently used for ILD-associated cough.
## Methods
**Strengths:**
- Clear inclusion/exclusion criteria for participant selection.
- Well-defined primary and secondary endpoints.
- Appropriate measurement tools (cough frequency, LCQ score, cough VAS) with consistent assessment intervals.
- Ethical approval and informed consent procedures properly documented.
**Weaknesses:**
- Small sample size (only 6 patients completed the study) limits statistical power.
- No placebo control group, making it difficult to distinguish treatment effects from natural variation or placebo effects.
- Short monitoring period for cough frequency (only 30 minutes), which may not capture variations throughout the day.
- Limited detail on the dosage of gefapixant administered to patients.
## Results
**Strengths:**
- Comprehensive presentation of patient characteristics including ILD subtypes, comorbidities, and concomitant medications.
- Multiple assessment parameters (objective and subjective) provide a more complete picture of treatment effects.
- Thoughtful analysis of interindividual variability, with clear visual representation in Figure 2.
- Honest reporting of non-statistically significant results.
**Weaknesses:**
- The small sample size severely limits statistical significance and generalizability.
- No subgroup analysis based on ILD type, which could have provided insights into differential effectiveness.
- Limited exploration of potential confounding factors that might explain the high interindividual variability.
- Some inconsistency between objective measures (cough frequency) and subjective assessment (LCQ score) not fully explained.
## Discussion
**Strengths:**
- Balanced interpretation of results, acknowledging limitations while still highlighting potential clinical relevance.
- Thoughtful consideration of the variability in treatment response.
- Good contextual placement of findings within existing literature on gefapixant.
- Appropriate discussion of taste disorder as a key side effect, including observations about its habituation over time.
**Weaknesses:**
- Could have more thoroughly discussed potential mechanisms explaining why some patients responded to treatment while others did not.
- Limited discussion of how these findings might influence clinical decision-making.
- Minimal exploration of how patient characteristics might predict treatment response.
- No cost-benefit analysis considering the side effects versus therapeutic benefits.
## Conclusion
**Strengths:**
- Appropriately cautious in claiming efficacy given the non-significant results.
- Acknowledges the need for further research to identify responsive patient subgroups.
- Highlights the potential of gefapixant despite limitations in the current study.
**Weaknesses:**
- Could provide more specific recommendations for future research directions.
- Lacks practical guidance for clinicians considering gefapixant for ILD patients with refractory cough.
## Overall Assessment
This study represents an important preliminary investigation into gefapixant for ILD-associated cough. Its primary strength lies in its real-world clinical approach and comprehensive assessment of both objective and subjective outcomes. The authors are commendably transparent about the study's limitations, particularly the small sample size and lack of statistical significance.
The paper's most significant contribution is highlighting the marked interindividual variability in treatment response, suggesting the need for a more personalized approach to managing ILD-associated cough. Future research with larger sample sizes, longer monitoring periods, and placebo controls will be essential to establish gefapixant's role in ILD treatment protocols.​​​​​​​​​​​​​​​​
Author Response
Response: We greatly appreciate your recognition of the study’s strengths, as well as the constructive comments regarding areas for improvement. Although no specific revisions were directly suggested, we have revised several areas of the manuscript based on comments from other reviewers. We hope that these revisions improve the clarity and completeness of the manuscript. Once again, we thank you for the helpful insights and balanced review.
Round 2
Reviewer 2 Report
Comments and Suggestions for Authors
Authors replied to my comments in a very satisfactorily way. Therefore, for me this paper can now be accepted.